# Orthoflavivirus Vaccine Platforms: Current Strategies and Challenges

**DOI:** 10.3390/vaccines13101015

**Published:** 2025-09-29

**Authors:** Giulia Unali, Florian Douam

**Affiliations:** 1Department of Virology, Immunology and Microbiology, Boston University Chobanian and Avedisian School of Medicine, Boston, MA 02118, USA; 2National Emerging Infectious Diseases Laboratories, Boston University, Boston, MA 02118, USA

**Keywords:** orthoflavivirus, orthoflavivirus envelope glycoprotein, vaccines, orthoflavivirus vaccine, live-attenuated vaccine, virus-like particle, DNA vaccine, mRNA vaccine, saRNA vaccine, adenoviral vaccine, viral vector vaccine, lentiviral vector vaccine, insect-specific flavivirus vaccine, multivalent vaccine

## Abstract

The Orthoflavivirus genus belongs to the *Flaviviridae* family. Orthoflaviviruses include major clinically relevant arthropod-borne human viruses such as Dengue, Zika, yellow fever, West Nile and tick-borne encephalitis virus. These viruses pose an increasing threat to global health due to the expansion of arthropod habitats, urbanization, and climate change. While vaccines have been developed for certain orthoflaviviruses with varying levels of success, critical challenges remain in achieving broadly deployable vaccines that combine a robust safety profile with durable immunity against many current and emerging orthoflaviviruses. This review provides a snapshot of established and emerging vaccine platforms against orthoflaviviruses, with a particular emphasis on those leveraging the envelope glycoprotein E as the primary antigen. We examine the strengths and disadvantages of these different platforms in eliciting safe, durable, and robust orthoflavivirus immunity, and discuss how specific attributes such as multivalency, authentic epitope presentations, and logistical practicality can enhance their value in preventing orthoflavivirus infection and disease.

## 1. Introduction to Orthoflaviviruses

Orthoflaviviruses are a genus within the *Flaviviridae* family. These viruses are transmitted by arthropods to humans, mainly via mosquitoes and ticks. With human infection ranging from asymptomatic cases to debilitating disease that can sometimes be fatal, orthoflaviviruses represent a significant health and economic concern worldwide [1]. In addition to the major, well-characterized orthoflaviviruses such as dengue virus (DENV), Zika virus (ZIKV), yellow fever virus (YFV) and West Nile virus (WNV), several other orthoflaviviruses are currently emerging in multiple parts of the globe. This includes Usutu virus in Europe, members of the tick-borne encephalitis virus (TBEV) serogroup in eastern Russia, and Powassan virus (POWV) in the Northeastern United States, Great Lakes regions and Canada [2,3,4,5,6,7,8,9]. As urbanization, habitat disruption, and climate change are likely to increase interactions between orthoflaviviruses and humans—mainly through the expansion of their arthropod vector endemic areas—there is a pressing need to develop effective therapeutic and prophylactic strategies to reduce their global health impact [1,10,11,12,13].

### 1.1. Orthoflavivirus Disease Spectrum

The spectrum of orthoflavivirus clinical outcomes is broad, ranging from silent carriage to life-threatening complications with no licensed antiviral therapeutics available. Most individuals experience no illness or only mild, flu-like symptoms characterized by fever, rash, headache, nausea, or muscle aches, which typically resolve within a week. However, a subset of patients develops severe or atypical disease hallmarks [14,15,16,17]. For instance, DENV infection can advance into hemorrhagic fever or shock syndrome, marked by intense capillary leakage, thrombocytopenia, and internal bleeding [18,19,20,21]. WNV infection, while asymptomatic in most individuals, can lead to neuroinvasive disease, including meningitis, encephalitis, and acute flaccid paralysis [22,23,24,25]. YFV and Japanese encephalitis virus (JEV) infection both frequently progress into visceral or neuroinvasive diseases. Yellow fever disease initially presents as a typical febrile illness but can escalate to hemorrhagic manifestations, jaundice, multi-organ failure, and up to 50% mortality among severe cases [26,27,28,29]. JEV causes encephalitis in 20–30% of infected individuals, with mortality rates near 30%, and survivors elicit a significant risk of permanent neurological impairment in survivors [30,31,32,33,34,35]. ZIKV stands out for its ability to breach the placental barrier [36,37]. While symptomatic infection typically involves mild fever, rash, arthralgia, and conjunctivitis, vertical transmission can lead to congenital Zika syndrome, characterized by microcephaly, brain malformations, and developmental delay in affected infants [38,39,40,41,42]. A small number of adults also experience neurological complications such as Guillain–Barré syndrome [43].

Members of the TBEV serogroup (Composed of European, Siberian and Far Eastern TBEV Subtypes) and POWV are neurotropic orthoflaviviruses transmitted by *Ixodes* ticks, both capable of causing severe and sometimes fatal encephalitis in humans [3,44]. TBEV infection often follows a biphasic course: the initial phase is characterized by non-specific flu-like symptoms such as fever, malaise, and myalgia, which may resolve before a second phase emerges in about 20–30% of symptomatic cases, involving central nervous system (CNS) manifestations such as meningitis, encephalitis, or myelitis [45,46,47]. In contrast, POWV infections typically present with an abrupt onset of CNS disease without a preceding febrile phase. Though many POWV infections are asymptomatic or mild, symptomatic cases can result in meningoencephalitis, seizures, and coma [44,48,49,50]. The case fatality rate for TBEV varies by subtype, with approximately 0.5–2% mortality in European and Far Eastern forms, respectively, and long-term neurological sequelae in up to 20% of survivors [51,52,53]. For POWV, the reported case fatality rate is significantly higher (~10–15%), with over 50% of survivors experiencing persistent neurological deficits such as paralysis, cognitive impairment, or chronic fatigue [2,54,55].

This diverse clinical profile across orthoflaviviruses, from mild and transient illness to severe hemorrhagic, neurological, and developmental disorders, reflects orthoflaviviruses’ different tissue tropisms and pathogenic mechanisms. The potential severity of orthoflaviviral disease underscores the necessity of vaccines capable of preventing both infection and critical clinical complications, particularly among vulnerable populations, including young children, the elderly and pregnant women [56,57,58,59,60].

### 1.2. Orthoflavivirus Epidemiology: A Growing and Global Threat

Over the past decade, the ecology and epidemiology of orthoflaviviruses have been significantly impacted by climate change, urbanization, and evolving vector ecologies [61,62,63,64,65,66]. DENV is the best example of this shift: in 2024, case numbers increased to an unprecedented 14.1 million globally, double the previous year’s count, with nearly 9500 recorded deaths, while early 2025 reports already show a 15% increase over the five-year average [67]. According to the World Health Organization (WHO), Latin America has been hit the hardest, with over 3.2 million infections and 1450 fatalities since January 2024, creating urgent pressure on public health infrastructure [67,68].

ZIKV also remains a persistent threat. By mid-2024, around 25,470 cases were confirmed in the Americas according to the Pan American Health Organization (PAHO). Cases were heavily concentrated in Brazil, alongside sporadic outbreaks in Southeast Asia. YFV too has expanded its reach. Alongside 61 confirmed cases and 30 deaths in the Americas in 2024 according to PAHO, the virus has recently re-emerged in São Paulo and Tolima, underscoring its rebound into urbanized areas.

JEV continues to impose a heavy toll with approximately 68,000 cases and 17,000 deaths annually, while new reports of cases in Australia hint at its increasing spread [69,70]. Simultaneously, WNV has begun appearing in unexpected geographical areas, such as in mosquito pools in the United Kingdom in July 2023 [71], and is becoming a growing threat in many European countries [8,72,73] along with USUV and TBEV [7,8,73,74,75,76,77].

Underlying this spread is the geographic expansion of mosquito and tick vectors. *Aedes aegypti* and *Ae. albopictus* are thriving in increasingly urban and temperate environments, with modeling predicting their establishment across much of Europe by 2040 and northern North America by mid-century [11,78]. Culex mosquitoes similarly benefit from warmer climates, extending the range of WNV [72,79]. Ixodes ticks, vectors for TBEV and Powassan virus, are also expanding into higher latitudes, introducing new pathogen risk zones [74,80].

Collectively, these observations suggest a shift in the global risk matrix. In the absence of effective therapies and with vaccines limited to a few orthoflaviviruses, there is an urgent need to develop robust, safe, and rapidly deployable vaccine strategies to prevent and contain emerging orthoflavivirus threats.

### 1.3. Orthoflavivirus Life Cycle and Antigenic Targets

Orthoflaviviruses are positive-sense, single-stranded RNA viruses with a genome of approximately 10–11 kb [1]. The orthoflavivirus replication cycle occurs entirely within the cytoplasm of infected cells. Upon entry into a host cell via receptor-mediated, clathrin-dependent endocytosis, the acidic environment of the endosome triggers conformational changes within the envelope (E) glycoprotein, inducing fusion of the viral membrane with the endosomal membrane and releasing the viral genome into the cytosol [81,82,83,84]. Owing to its receptor-binding and fusion roles [85,86,87,88,89], E is the primary target of neutralizing antibodies, making it a crucial antigen for vaccine design [90]. Composed of three domains (DI–DIII), with the fusion loop localized in DII and receptor-binding sites in DIII, E protein glycosylation (at residues such as Asn130, Asn175, Asn207) fine-tunes tissue tropism and immunogenicity [91,92,93,94].

Following viral entry, the viral genome is translated into a single polyprotein, which is then co- and post-translationally cleaved into three structural proteins, capsid (C), the pre-membrane/membrane (prM/M) protein and E, and seven non-structural proteins (NS1–NS5) (Figure 1A). While C, PrM/M and E shape the virion structure [95,96], non-structural proteins orchestrate genome replication, immune evasion, and virion assembly [97,98,99,100,101,102]. Newly synthesized RNA molecules, via the activity of the RNA-dependent RNA polymerase (RdRp) NS5, complex with capsid proteins [103] to form viral capsids. ER membrane-anchored E-prM trimers then promote curvature of the ER membrane around viral capsids to form “spiky” immature particles [104,105]. Particle migration to the trans-Golgi network (TGN), a mildly acidic environment, rearranges E-prM trimers to form particles displaying 90 E homodimers organized in a smooth conformation, simultaneously exposing a furin cleavage site (FCS) within prM [106,107,108,109,110]. Furin, a resident TGN protein, then cleaves prM into M and a pr peptide [108,109]. An N-terminal, “globular,” pr peptide remains associated with E, shielding the E fusion loop domain and preventing premature fusogenic rearrangement of the E dimers [107,108,109,111]; While the C-terminal, membrane-anchored part of prM (i.e., M) sits below the E dimers.

Upon exocytosis at neutral pH, a localized conformational change within E DI releases the pr peptide, yielding mature infectious virions [111,112,113,114] (Figure 1B,C). However, incomplete furin cleavage and the required metastability of E dimers drive structural heterogeneity among viral progenies, which has implications for viral entry and humoral responses [95,96]. Particularly, incomplete furin cleavage yields immature virions that are not infectious, due to pr blocking interactions between the E fusion loop and the endosomal membrane at acidic pH and the masking of the FCS at neutral pH [107,108,109,111] (Figure 1C).

Although prM ensures proper E oligomeric assembly [81,95,111,115], the prM/E-dependent particle maturation process can be variable between mosquitoes and tick-borne orthoflaviviruses. In YFV E, an internal DI loop occludes the pr-binding pocket on the E dimer at neutral pH, so pr cannot bind the dimer (though it binds E monomers) [114]. At TGN pH, this loop rearranges to expose this site, allowing pr to cap the fusion loop and block premature fusion. This is in contrast to TBEV, where pr binds to E dimers at neutral pH, allowing for the stabilization of E dimers at acidic pH [113]. Tick-borne orthoflavivirus immature particles, unlike mosquito-borne orthoflaviviruses, also remain fully infectious because of the irreversible exposure of the FCS at neutral pH, which enables furin-mediated cleavage of prM at the surface of target cells upon viral entry [116].

By promoting E oligomerization and shaping virion structural heterogeneity, prM is pivotal for exposing clinically relevant E epitopes, underscoring its relevance for vaccine design [117]. However, its inclusion in vaccine constructs also raises concerns over antibody-dependent enhancement (ADE) via non-neutralizing antibody responses [118,119], prompting modern vaccine platforms to favor truncated prM or stabilized E-only immunogens to mitigate ADE risks [120,121,122,123].

Beyond prM and E, the non-structural protein NS1 has emerged as a valuable adjunct antigen in vaccine design [124,125,126]. NS1 exists as a membrane-bound dimer and a secreted hexamer; its glycosylation-driven assembly and lipid-rich structure facilitate immune evasion through antagonism of complement pathway mediators, including binding factor H and C4 regulatory proteins [127]. NS1 also supports viral RNA replication by interacting with NS4A/B in ER-derived membranes, while its secretion contributes to pathogenesis [127,128,129]. Immunologically, NS1 elicits a multifaceted protective response [127,130]. Antibodies raised against NS1 can inhibit endothelial cell dysfunction, trigger antibody-dependent cellular cytotoxicity (ADCC), and reduce viral titers without inducing ADE, an attribute that enhances the antigen’s appeal in multi-component vaccines [124,130,131]. Experimental vaccines incorporating NS1 into both protein and DNA platforms have demonstrated broader protection against DENV and WNV in animal models, underscoring its potential to reinforce immune breadth [124,130,131].

## 2. Orthoflavivirus Vaccine Platforms: From Historical to Next-Generation Approaches

Orthoflavivirus vaccine research has evolved remarkably, progressing from traditional live-attenuated viruses to nucleic acid-based platforms, such as mRNA vaccines. The section below provides a brief overview of the benefits and limitations of different orthoflavivirus vaccine platforms, with a specific focus on those leveraging E (except Section 2.7) as the primary antigen (Table 1; Figure 2).

### 2.1. Live Attenuated Vaccines

Live-attenuated vaccines and their chimeric derivatives remain currently the dominant approach of orthoflavivirus immunization, delivering potent, long-term protection that closely resembles natural immunity [90,132]. The yellow fever vaccine (YFV-17D) and the Japanese encephalitis SA14-14-2 strains are exemplary in their ability to elicit robust, lifelong antibody and T-cell responses with a single dose [133,134,135,136]. By leveraging the YFV-17D backbone fused with the envelope proteins of other orthoflaviviruses, chimeric vaccines such as Dengvaxia and IMOJEV extend this protective model to DENV, ZIKV and JEV [133,137,138,139]. However, this has been associated with significant safety concerns.

For example, Dengvaxia has been linked to vaccine-enhanced disease in seronegative individuals due to the risk of ADE [137,140,141].

In contrast, the next-generation TAK-003 (Qdenga) employs a Dengue-2 backbone to deliver prM-E from all four dengue serotypes [142,143]. This achieves balanced immunogenicity regardless of prior orthoflavivirus exposure as demonstrated in phase III trials with sustained efficacy (~80% protection) and no increase in hospitalizations among seronegative recipients [142,143,144]. Notably, neither TAK-003 nor its predecessors offer heterologous protection against other orthoflaviviruses such as ZIKV or WNV, and their ADE potential in cross-flavivirus scenarios remains largely untested [90]. Pre-existing orthoflavivirus immunity can also skew responses and predispose to off-target enhancement, which underscores the urgent need for multivalent or pan-flavivirus vaccines.

While live-attenuated strategies elicit potent immunogenicity and durable protection, they still have several limitations. For instance, their cold-chain dependence and logistical challenges associated with large-scale production restrict accessibility and scalability [145,146]. Their replication competence also precludes their use in immunocompromised individuals or during pregnancy [147]. YFV-17D vaccination can result in rare events of vaccine-associated viscerotropic diseases [148], which have been associated with inherited defects in type I interferon (IFN) signaling [149]. This, coupled with risks of ADE (particularly in the context of DENV, WNV, and ZIKV monovalent vaccination [150,151,152]), represents a major challenge for the large-scale implementation of novel orthoflavivirus live-attenuated vaccines (Table 1).

### 2.2. Inactivated Vaccines

Inactivated vaccines are a long-standing, yet still used, platform for orthoflavivirus prevention. These vaccines employ whole virions inactivated through chemical (e.g., formalin, β-propiolactone) or physical methods. This strategy can preserve native antigenic structures, most notably the envelope glycoprotein (prM-E), while abrogating infectivity [153,154]. Unlike live-attenuated alternatives, inactivated vaccines cannot replicate or revert and display low to no safety concerns in immunocompromised individuals and pregnant women.

Interest in inactivated orthoflavivirus vaccines has surged in recent years. For instance, DENV-purified inactivated tetravalent candidates (TDENV-PIV) have elicited strong neutralizing antibody responses in phase I/II trials, setting a precedent for safe, multivalent formulations [155]. Notably, research on ZIKV vaccines pivoted toward inactivated approaches culminating in two independent Phase I trials of ZPIV (formalin-inactivated PRVABC59 strain), which confirmed both its immunogenicity and safety profile [156,157].

Recent progress in inactivated WNV vaccines has also been promising. The hydrogen peroxide–inactivated HydroVax-001 candidate advanced through Phase I trials with a strong safety profile and elicited neutralizing antibodies in 31–50% of recipients [158]. Additionally, a formalin-inactivated whole-virion vaccine demonstrated excellent immunogenicity in preclinical models, achieving 100% seroconversion and full protection in mice [153,159].

Inactivated whole-virus vaccines serve as the gold standard for preventing infections by members of the TBEV serogroup. Formalin-inactivated whole-virus formulations derived from the European TBEV subtype represent the primary preventive strategy in endemic regions across Europe and Asia [160,161,162,163,164]. Notable examples include FSME-Immun, Encepur, and Ticovac. Large-scale observational studies and clinical trials have demonstrated that these vaccines are both highly immunogenic (seroconversion in >90% of recipients) and effective, with vaccine efficacy estimated above 90% in preventing clinical disease [160,161,162,163,164].

Although inactivated vaccines offer several advantages, their inability to replicate reduces cellular responses, especially CD8^+^ T-cell responses. Consequently, these vaccines often require the use of adjuvants and multiple doses to achieve robust protection, increasing logistical complexity and cost [165,166]. Moreover, the inactivation process itself can disrupt key conformational epitopes, resulting in reduced efficacy, while inadequate inactivation poses safety hazards [153]. ADE concerns from monovalent vaccine formulations also remain similar to live-attenuated vaccines, and may be further amplified if the inactivation strategy affects specific epitopes. Scalable production also demands high-biosecurity facilities capable of manufacturing, inactivating, and formulating large quantities of live virus, which can delay production during outbreaks (Table 1).

### 2.3. Nucleic Acid Vaccines: Messenger RNA (mRNA) and Self-Amplifying (saRNA)

The advent of nucleic acid vaccines has transformed our approach to orthoflavivirus prevention by offering unparalleled adaptability and safety. These platforms bypass traditional challenges inherent to live-attenuated and inactivated vaccines, allowing swift adaptation against emerging strains (Table 1).

#### 2.3.1. mRNA and DNA Vaccines

mRNA vaccines have emerged as a powerful strategy in preventing infectious diseases, and orthoflaviviruses are no exception [167,168]. Unlike more traditional platforms, mRNA vaccines are modified nucleoside-encoded viral antigens formulated into lipid nanoparticles (LNP) [169,170]. Following delivery into host cells, vaccine antigens are translated from mRNAs without significant induction of IFN responses. Antigens can be secreted or not, and can elicit both humoral and cellular responses [167,168,169,170]. While COVID-19 demonstrated the unique clinical potential of mRNA vaccines, the foundational breakthrough was earlier, through the discovery that incorporating specific modified nucleotides (e.g., N1-Methylpseudouridine) into mRNA prevents the induction of deleterious type I IFN responses [171].

Preclinical testing of orthoflavivirus mRNA vaccines has yielded promising results. A DENV-1 prM/E mRNA vaccine yields strong neutralizing antibodies and cellular immunity in immunocompetent mice and protects mice from fatal disease [172]. Building on this, multivalent formulations targeting all four DENV serotypes, including the use of NS1 and epitope-engineered E domain III, have delivered promising results in mouse models, offering broad protection with minimal risk of ADE [173]. PrM-E-encoded mRNA vaccines against POWV and ZIKV have also demonstrated strong efficacy in mouse models [174,175,176] and non-human primates [177]. Early human trials of mRNA-1325 and mRNA-1893, two ZIKV mRNA vaccines encoding for prM-E of different ZIKV strains, demonstrated that the latter achieved potent and durable neutralizing responses that endured for up to one year and conferred sterilizing protection in non-human primates [177,178].

mRNA vaccines have several advantages over other existing platforms. The manufacturing process is fast and highly scalable, providing flexibility in dosing and antigen composition [179]. This makes this platform ideal for responding to emerging variants or new orthoflaviviruses. Moreover, the absence of live viruses eliminates concerns about reversion. Nonetheless, mRNA vaccines are poorly stable and have a relatively short half-life, which leads to limited durability of systemic humoral responses [179]. They also require stringent cold-chain storage, which complicates distribution, especially in warm, outbreak-prone regions where reliable refrigeration is scarce [180]. LNP-associated reactogenicity can also contribute to adverse reactions upon vaccination [181,182].

DNA vaccines represent a potential alternative to the low stability of mRNA vaccines, in addition to being more versatile in terms of genetic manipulations. A ZIKV DNA vaccine expressing prM and E was shown to be protective against infection in preclinical animal models [183], and to induce neutralizing responses in a Phase I clinical trial [184]. A veterinary DNA vaccine expressing prM and E of WNV can prevent WNV infection in mice and horses [185], and was licensed for horse vaccination in the United States in 2005. Phase I clinical trials have shown that DNA plasmids encoding prM and E under the control of a CMV or modified CMV/R promoter are safe and induce neutralizing responses [186,187]. Despite their stability advantage, DNA vaccines are limited by the large DNA dose requirement compared to mRNA (1–4 mg/injection) and most notably by the need for electroporation at the site of inoculation to enhance immunogenicity.

A common challenge faced by mRNA and DNA vaccines also lies in mimicking clinically relevant prM-E oligomerization. prM-E-encoded mRNA and DNA vaccines induce the spontaneous secretion of ~30 nm subviral particles (SVP) harboring heterogeneous levels of prM-E and M-E at their surface [175]. The absence of a viral capsid and the heterogeneity of prM cleavage, which depends on the cells and tissues targeted, may result in suboptimal epitope exposure. This is also associated with ADE risks, similarly to live-attenuated and inactivated vaccines [188,189]. While this could be mitigated by the design of multivalent cocktails of mRNA or DNA vaccines, balancing the expression and pharmacokinetics of the different antigens in a consistent fashion would represent a considerable challenge, in addition to significantly complicating the manufacturing process.

Collectively, while genetically based vaccines offer scalability and rapid manufacturing potential, their full potential in orthoflavivirus immunization will depend on overcoming challenges around the durability of sterile immunity, the induction of clinically relevant memory responses, thermostability (for mRNA vaccines), and delivery (for DNA vaccines) (Table 1).

#### 2.3.2. Self-Amplifying RNA (saRNA) Vaccines

Self-amplifying RNA (saRNA) vaccines are poised to redefine our approach to orthoflavivirus immunization. Unlike conventional mRNAs, saRNA constructs carry both the antigen sequence, typically prM-E, and genes encoding a viral replicase that are often derived from alphaviruses [190]. Once delivered via lipid nanoparticles or nanostructured carriers, the replicase amplifies the RNA in situ and drives strong and sustained antigen production even with a reduced dose [191,192]. Recently, the first saRNA vaccine was licensed in Japan. This saRNA vaccine promotes robust protection against COVID-19 [193,194] and, most notably, induces more durable persistence of serum neutralizing antibodies compared to mRNA vaccines [195], highlighting a key benefit of saRNA over mRNA vaccines.

Notably, in preclinical orthoflavivirus models, saRNA demonstrates remarkable potency [191,192,196,197]. For instance, a saRNA vaccine encoding ZIKV prM-E elicited robust antibody and CD8^+^ T-cell responses in mice after a single microgram dose [191,197,198]. This saRNA induced significant protection against ZIKV challenge, although durability waned by day 84 after vaccination [191]. Co-administration of T-cell co-stimulatory agonists enhanced long-term immunity, demonstrating the platform’s flexibility.

Although saRNA constructs are larger than mRNA vaccines (9–11 kb), they have shown compatibility with nanoparticle delivery systems [199,200,201]. Notably, vaccine formulations such as saRNA–Nanostructured Lipid Carrier (NLC), a thermostable lipid-based delivery platform for RNA vaccines, can be lyophilized and stored at 4 °C for over 21 months, offering good stability at room temperature for months [202,203].

Even though saRNA vaccines have shown promise in preclinical animal models, their clinical translation has historically been underwhelming. This is mainly due to the induction of elevated type I IFN responses and inflammation upon RNA replication and double-stranded RNA (dsRNA) formation, which hampers antigen production, enhances cytotoxicity, and the induction of effective adaptive immunity [204,205,206,207,208,209]. The recent discovery that specific modified nucleotides, such as 5-methylcytosine (m5C), are compatible with RdRp saRNA activity and enable escape from type I IFN responses has re-ignited the potential of saRNA to serve as a powerful vaccine platform for clinical application [210]. Specifically, m5C-saRNA vaccines have been shown to mediate superior protection against fatal SARS-CoV-2 infection at a 10 ng dose compared to non-modified nucleotide saRNA and mRNA vaccines [210]. Particularly, this discovery opens avenues for developing multivalent m5C-incorporated saRNA vaccines that induce high levels of expression of three to four different antigens from a single saRNA molecule. Beyond nucleotide modifications, researchers have also introduced strategic mutations, such as alterations in the macrodomain of alphavirus nsP3, which dampen dsRNA detection, enhance translation efficiency, and reduce cellular toxicity [211].

Beyond increased and prolonged antigen expression, a significant value of saRNA vaccines lies in their ability to more potently activate innate immune responses as compared to mRNA through their self-amplifying ability. However, this means that enhancing the immune escape properties of saRNA vaccines could also paradoxically reduce their efficacy. Nucleotide modifications such as m5C are well-positioned to achieve this complex balance, as they are only incorporated into the initial (i.e., inoculated) saRNA molecules. Upon saRNA replication, novel saRNA progenies only incorporate wild-type non-modified nucleotides, which can then form reactive dsRNA molecules that activate innate immune responses. These observations suggest that early escape of modified-nucleotide saRNAs (i.e., at the time of cell entry) from pattern recognition receptors is sufficient to enhance antigen production and immunogenicity over non-modified saRNA and mRNA vaccines [210]. This has been further exemplified by evidence that early and transient inhibition of type I IFN responses can enhance the efficacy of non-modified nucleotide saRNA vaccines [212].

m5C has opened the way to the rational development of saRNA vaccines that trigger balanced levels of innate immune responses to maximize vaccine immunogenicity, but significant work remains. Given the current paradigm that non-nucleotide modified saRNA vaccines underperform in clinical settings due to the high level of inflammation they induce [204,205,206,207,208,209], a particular challenge will be to measure how modified-nucleotide saRNA vaccines, and the different levels of innate immune activation they drive, perform in individuals with different immune history and/or inflammatory baselines (e.g., in individuals with inflammatory disorders). Furthermore, although saRNA platforms offer promise for the development of multivalent vaccines expressing three to four orthoflavivirus antigens on the same molecules, the dynamics of antigen expression (e.g., the order in which the antigens are positioned in the genetic cassette) could significantly skew antibody responses across targets. As this could have indirect consequences on ADE risks, significant work will be required to identify optimal antigen expression cassettes that maximize neutralizing responses against multiple flaviviral targets while mitigating ADE risks.

While some evidence suggests that saRNA vaccines may be superior to mRNA vaccines in that aspect [195], the durability of immunity they induce still requires optimization. Preclinical studies have shown that saRNA can elicit strong initial antibody and T-cell responses from a single administration, but these responses often wane within a few months if not reinforced by booster doses or immunostimulatory adjuvants [191]. To achieve lasting protection, saRNA-based vaccine strategies may need to be combined with heterologous boosters, such as protein- or vector-based vaccines. Moreover, the manufacturing and regulatory demands for saRNA vaccines are substantial. The replicon’s large size complicates production under Good Manufacturing Practices (GMP), including higher quality control and regulatory complexity compared to conventional mRNA [205]. Lastly, like canonical mRNA strategies, saRNA vaccines face challenges related to LNP reactogenicity and improper folding of the prM-E complex onto SVPs, potentially undermining clinically relevant immunogenicity.

In summary, while saRNA presents an unparalleled tool for inducing potent immunity at low doses, its path forward hinges on sophisticated molecular engineering to balance innate immune activation, sustain durable immune memory, and streamline scalable manufacturing (Table 1).

### 2.4. Virus-like Particles (VLPs) Vaccines

Virus-like particles (VLPs) have emerged as a practical platform for orthoflavivirus vaccination. Producing orthoflavivirus VLPs-based vaccines typically involves co-expression of prM and E proteins in mammalian, insect, yeast, or plant cell systems [213,214,215,216,217]. Transfection of prM- and E-coding DNA constructs into bioreactor cell lines drives the self-assembly of prM-E heterodimers incorporated into the membrane of VLPs, which bud within the endoplasmic reticulum and are secreted extracellularly [218]. Host furin-mediated cleavage of prM in the Golgi converts these into VLPs that mimic the viral architecture [219,220], exhibiting small (~30 nm) or full virion-like (~50 nm) morphologies [215].

While similar in nature, VLP and SVP significantly differ in the way they are produced, i.e., in bioreactor cell lines (ex vivo) versus in the vaccine recipient’s cells targeted by mRNA/saRNA formulations (in vivo). These differences in particle production processes are likely to influence particle size, the degree of maturation (i.e., the level of prM cleavage), and glycan/lipid composition, all of which are likely to affect immune sensing and immunogenicity. It is worth noting that SVPs are also more likely to drive direct MHC-I antigen presentation (through initial prM-E endogenous expression), as compared to exogenous E incorporated onto VLPs.

VLPs are highly immunogenic in preclinical models, eliciting high titers of neutralizing antibodies and generating persistent memory T-cell responses against DENV and ZIKV [214,215,221,222,223,224,225]. Similar to genetic vaccines, VLPs deliver a valuable balance between mimicking the structural authenticity of orthoflavivirus particles and mitigating risks associated with live viral pathogens in the vaccination process (i.e., live-attenuated vaccines) or during manufacturing (i.e., inactivated vaccines).

VLP yield and quality are significantly influenced by production techniques [226,227,228]. Several studies have reported improved secretion and assembly of VLPs through codon optimization, inclusion of native signal peptides and promoters, and the adoption of low-temperature culture (for example, 28 °C) in mammalian systems [229]. Moreover, the flexibility in selecting the expression hosts, whether yeast (e.g., *Pichia pastoris*), insect (baculovirus), mammalian cell lines, or plant systems, allows tailoring of glycosylation patterns, scalability, and processing costs [227,230]—unlike mRNA and DNA vaccines. Early trials with yeast-derived VLPs show they can be produced cost-effectively without compromising immunogenicity [228].

However, standardizing and controlling optimal envelope maturation and glycosylation remain significant challenges for VLPs to advance in clinical applications (Table 1). Low-titer production also poses substantial manufacturing challenges for large-scale production. Mitigating ADE risks also requires significant molecular engineering and quality control efforts during the VLP production process. Additionally, designing multivalent VLP cocktails to limit ADE also poses significant manufacturing challenges. VLP vaccines also often require inclusion of adjuvants or multiple doses to achieve long-lasting immunity [231,232,233], collectively further complexifying manufacturing and quality control procedures.

### 2.5. Insect-Specific Flaviviruses (ISFVs) Vaccines

Insect-specific flaviviruses (ISFVs), such as Binjari (BinJ) virus and Aripo virus, are naturally restricted to replication in arthropods and are incapable of infecting vertebrate cells [234,235]. This vertebrate-replication block, combined with a capacity to tolerate envelope gene swaps from pathogenic orthoflaviviruses, makes them an ingenious and safe backbone for orthoflavivirus vaccine development.

ISFV chimeras are generated by replacing the prM-E structural genes of a vertebrate-infecting orthoflavivirus (e.g., DENV, ZIKV, YFV, WNV) with those from an ISFV using methods such as Circular Polymerase Extension Reaction (CPER) [236,237]. Notably, these chimeras replicate to very high titers (up to ~10^9^·^5^ CCID_50_/mL or ~7 mg/L) in mosquito cell lines like C6/36, yet remain entirely replication-defective in vertebrate cells [234,235,236,237], yielding an intrinsically safe profile and enabling production under minimal biosafety levels [238,239].

ISFV chimeras promote strong neutralizing antibodies while also eliciting robust CD4^+^ and CD8^+^ T-cell responses [240,241]. ISFV-prM/E chimeras have generated notable preclinical results. For instance, BinJ/WNVKUN-prME vaccine induced strong neutralizing antibody responses and provided complete protection in mice against lethal WNV NY99 challenge [242]. Another example is the BinJ/Zika-prME vaccine, which, with a single dose, conferred protection in IFNAR^−^/^−^ mice that persisted for at least 14 months [243]. Notably, in a dual-orthoflavivirus approach, bivalent ISFV chimeras combining prM-E genes from multiple orthoflaviviruses have demonstrated effective multivalent immunogenicity with no evidence of pathogen interference [237].

Despite these advantages, two major hurdles remain. First, the use of mosquito cell substrates complicates regulatory approval. Unlike cell lines such as VeroE6, CHO, or HEK293, C6/36 cells are not certified for the production of GMP-grade human vaccines. Furthermore, residual mosquito proteins, even from a virus-free culture, have, in human trials, caused immediate or delayed-type hypersensitivity reactions upon intradermal or subcutaneous administration [244,245]. Second, vaccine structural fidelity warrants close attention. E folding at the orthoflavivirus particle surface is temperature-dependent. Unlike the bumpy particle surface observed at 37 °C, orthoflavivirus particles display a smooth surface at 28 °C, at which mosquito cells are cultured [246,247]. Furthermore, differential glycosylation between human and insect cells may also affect prM-E epitope presentation [248]. While cryo-EM and monoclonal antibody binding have shown high structural similarity for BinJV-prM-E constructs [249], even subtle conformational deviations could influence immunogenicity.

Overall, ISFV chimeras combine the safety of VLP with enhanced structural authenticity. However, their large-scale implementation as vaccines is hindered by regulatory roadblocks associated with insect cell-based production, risks associated with allergenic contaminants, and their ability to present a breadth of clinically relevant viral epitopes (Table 1).

### 2.6. Viral Vector Platforms

Viral vector platforms, including adenoviruses (Ad) and lentiviral vectors (LV), leverage the natural infection pathway of replication-deficient mammalian viruses to express orthoflavivirus antigens in vivo. These vectors combine structural protein expression with targeted delivery, while demonstrating a robust safety profile (Table 1).

#### 2.6.1. Adenovirus Vector Vaccines

Replication-defective adenoviral vectors, such as human Ad5 and Ad26, as well as chimpanzee-derived ChAdOx1, have been successfully exploited to deliver DNA genome-encoded antigens, eliciting both potent CD8^+^ T-cell responses and helper T-cell-driven antibody production [250,251,252,253,254].

An Ad5–vector encoding for the full-length ZIKV prM-E triggers robust neutralizing antibody titers and a strong CD8^+^ T-cell response, leading to complete protection against lethal Zika challenge in preclinical mouse models [255]. Meanwhile, Ad4-prM-E, another human serotype, elicits protective T-cell immunity alone, underscoring the capacity of adenoviral vectors to confer disease resistance even in the absence of detectable antibodies [255]. Another compelling case is the chimpanzee-derived ChAdOx1-ZIKV, engineered with a prM-E cassette lacking the envelope’s transmembrane domain [256]. With a single unadjuvanted dose, this vector induces high neutralizing antibody levels and robust T-cell responses in both mice and non-human primates, while importantly showing no evidence of antibody-dependent enhancement [256]. The safety and immunogenicity of an Ad26 vaccine against ZIKV were also assessed through phase I clinical trials, which showed the induction of robust humoral and cellular responses without major safety concerns [257].

However, a notable limitation of this platform remains the pre-existing immunity to common human adenovirus serotypes (e.g., Ad5) [258,259,260,261], which can blunt vaccine efficacy in populations with prior exposure. Limitations associated with E epitope presentation onto SVPs also remain similar to those of several other vaccine platforms, including genetic vaccines and VLPs. The COVID-19 pandemic also highlighted efficacy, durability and safety issues associated with Adenovirus vaccines in humans [262,263,264,265], which will remain to be addressed.

Overall, while adenoviral vector versatility and immunogenic properties make them attractive candidates in the development of next-generation orthoflavivirus vaccines, improving their durability and safety profiles will be key for their large-scale implementation (Table 1).

#### 2.6.2. Lentiviral Vector Vaccines

Lentiviral vectors (LVs) harboring an RNA-based transgene encoding for WNV prM-E present a compelling platform in orthoflavivirus vaccinology. In murine models, a single intramuscular injection of an LV expressing WNV prM-E elicits high-titer neutralizing antibodies and vigorous T-cell responses, offering complete protection against lethal challenge [266]. Remarkably, antigen expression remains detectable for over a month, far exceeding the transient kinetics typical of mRNA vaccines.

A similar strategy was employed to vaccinate pigs with lentiviral vectors (TRIP/JEV) encoding the native JEV prM-E polyprotein [267]. When administered intramuscularly in pigs (domestic piglets), two doses of TRIP/JEV vectors induce high titers of anti-JEV IgG and strong in vitro neutralization activity across genotypes 1, 3 and 5 JEV. This was further confirmed in mice. However, the integration of lentiviral vector genomes into host chromosomes results in prolonged antigen expression and persistent antigen presentation, an undesirable feature for most vaccine applications, since continual antigen exposure can promote tolerance or T-cell exhaustion rather than optimal memory responses.

Integrase-defective lentiviral vectors (IDLVs) are well-positioned to address this issue as they combine durable antigen expression without the risks associated with genome integration. Their episomal persistence in antigen-presenting cells not only sustains germinal center activity but also favors the development of high-quality B-cell and T-cell memory [268,269,270,271,272,273]. In line with these findings, a non-integrating lentiviral vector (NILV) encoding a consensus ZIKV prM-E antigen, delivered as a single intramuscular dose, induces high-titer neutralizing antibodies and sterilizing protection against ZIKV infection across both immunocompetent and immunodeficient mouse models, with efficacy evident as early as 7 days post-immunization [274]. IDLVs were also engineered to express a secreted form of WNV E (sE) from the virulent IS-98-ST1 strain. A single, ultra-low dose of a non-integrative lentiviral vector encoding WNV sE elicits early (≤7 days) neutralizing antibody responses in mice. These responses result in sterilizing immunity against a lethal WNV challenge [275]. Notably, protection was durable (≥90 days)without the use of adjuvants or booster doses, demonstrating that episomal lentiviral platforms can elicit persistent humoral immunity and confer long-term protection via a single dose, characteristics highly desirable for emergency-use orthoflavivirus vaccines.

Unlike human adenovirus, IDLVs evade pre-existing anti-vector immunity, enabling multiple administrations without diminishing efficacy [272,273]. Compared to mRNA platforms, IDLV responses develop more gradually, yet maintain elevated antibody and T-cell levels for longer periods [271,276]. In contrast to VLP approaches, which deliver rapid humoral responses but often require adjuvants and boosters, IDLVs offer prolonged stimulation of both arms of the adaptive immune system from a single dose [271,276].

However, similarly to other platforms, IDLV-mediated expression of E and its subsequent incorporation onto VLP can result in suboptimal epitope presentation as discussed above. Despite the potential of the lentiviral core to foster authentic E oligomerization and yield particles with uniform architecture, efforts to pseudotype IDLVs with prM-E have been unfortunately unfruitful and plagued by low production yield. Significant work is needed to achieve the production of high-yield pseudotyped IDLV compatible with GMP-grade scales.

Despite these barriers, prolonged antigen expression, balanced induction of humoral and cellular immunity, and minimal safety risks underscore IDLVs as a valuable vaccine platform, particularly in single-dose or heterologous prime-boost regimens (Table 1).

**Table 1 vaccines-13-01015-t001:** Summary of the different flavivirus vaccine platforms discussed in this review.

Platform	Examples	Advantages	Disadvantages	Development Stage	Ref.
**Live** **Attenuated** **Vaccines**	YFV-17D (YFV), SA14–14–2 (JEV)	Long-lasting immunity; strong humoral and cellular responses; single dose; Preservation of antigenic structure	Safety concerns; ADE risks; Cold-chain requirements	Several licensed (e.g., YFV-17D, TAK-003)	[133,134,135,136,137,138,139,142,143,144]
**Inactivated vaccines**	Ixiaro (JEV), Ticovac (TBEV)	Safe for vulnerable individuals; No viral replication; Preservation of particle structure	Limited immunity; Requires adjuvant and multiple doses; Potential ADE risks; Potential epitope disruption	From Phase I/II trials (TDENV PIV; HydroVax 001) to licensed (Ixiaro)	[153,154,155,156,157,158,159,160,161,162,163,164]
**mRNA vaccines**	mRNA-1893 (ZIKV prM–E)	Rapid design and deployment; Potent systemic immunity	Limited durability of serum neutralizing responses; Cold-chain requirements; Authentic epitope presentation; ADE risks	From preclinical to Phase I/II (Zika mRNA-1325 and 1893)	[172,174,175,176,177,178]
**DNA vaccines**	GLS-5700 (ZIKV prM-E DNA plasmid)	Thermostability, versatility in genetic modifications	Low immunogenicity; High doses for injection; Authentic epitope presentation; Electroporation needed to enhance immunogenicity	From preclinical to Phase I trials	[183,184,185,186,187]
**saRNA vaccines**	Bivalent saRNA (ZIKV + YFV prM-E)	High antigen expression; More durable serum neutralizing responses compared to mRNA vaccines; Self-replication enhances adaptive immune activation; Multivalence and single-dose potential	Need to balance activation and evasion of cell-intrinsic immunity; Manufacturing challenges; ADE risks if monovalent; Authentic epitope presentation	Preclinical	[191,192,196,197,198]
**VLPs**	VLPs ZIKV prM-E	Presentation of authentic, membrane-anchored E protein; strong safety profile; potent immune responses	Manufacturing challenges because of low-titer production; ADE risks; require boost; Authentic epitope presentation	Preclinical	[214,215,221,222,223,224,225]
**ISFVs**	BinJ/WNVKUN-prME (WNV prM-E)	Grows to high titer in insect cells; Non-replicative in vertebrates (safe); Strong humoral and cellular responses	Requires insect-cell production; no GMP production capabilities	Preclinical	[237,240,241,242,243]
**Adenoviral Vectors**	ChAdOx1-ZIKV; Ad26.ZIKV.001	Potent humoral and cellular responses with single dose with no evidence of ADE in animal models	Safety concerns; Authentic epitope presentation; Durability and efficacy concerns in humans; Pre-existing immunity	From preclinical to Phase I (Ad26.ZIKV.001)	[255,256,257]
**Lentiviral Vectors**	TRIP/sE_WNV (WNV prM-E)	Single-dose sterilizing immunity in mice; potent neutralizing antibodies	Integration into host genome; manufacturing complexity; Authentic epitope presentation	Preclinical	[266,267,274,275]

### 2.7. Anti-Vector Vaccines

Anti-vector vaccines, also known as transmission-blocking vaccines, aim to block the pathogen transmission cycle by priming an immune response against arthropod proteins associated with viral pathogen transmission at the bite site. A notable advantage of such a strategy is its ability to target any pathogens transmitted by arthropods, whether known or unknown. A notable example is the 64TRP vaccine, which utilizes a recombinant protein derived from *Rhipicephalus appendiculatus* ticks [277]. Mouse immunization with specific modalities of the 64TRP vaccine reduced viral transmission from infected tick vectors to mice and vice versa, and decreased the incidence of fatal infection to a similar extent as a licensed TBEV vaccine [278]. Similar anti-vector approaches targeting mosquito proteins (e.g., salivary or midgut proteins) are being explored [279,280,281,282,283], including through Phase I clinical trials [284]. While anti-vector vaccines represent promising complementary strategies to conventional orthoflavivirus immunization, a detailed description of these platforms is beyond the scope of this review and is comprehensively covered in other reviews [285,286].

## 3. Orthoflavivirus Multivalent Vaccine Platforms

Multivalent vaccines targeting multiple orthoflaviviruses (DENV, ZIKV, WNV, JEV) are valuable to mitigate ADE concerns while facilitating the global health management of orthoflavivirus threats. However, their design poses considerable challenges due to antigenic interference and manufacturing complexity.

One promising approach uses tetravalent E-dimer VLPs, which display all four DENV serotype antigens on a single nanoparticle scaffold [287,288]. Recent data suggest that liposome-anchored E-dimer formulations can minimize immune interference while eliciting potent neutralizing responses to all dengue serotypes and could be easily adapted to include Zika or WNV antigens [289,290]. Similarly, VLP-based cocktails have shown promising preclinical results. For example, stable cell lines secreting VLPs from multiple orthoflaviviruses, including DENV, ZIKV, WNV, JEV and YFV, have generated high-titer neutralizing antibodies against all included antigens in mice, and large-scale suspension culture systems make them suitable for manufacturing [291].

On the nucleic acid front, multivalent saRNA or mRNA formulations are being explored [173,196]. A bivalent saRNA vaccine targeting both YFV and ZIKV induces high-titer neutralizing antibodies and multifaceted T-cell responses in mice and hamsters [196]. Importantly, this strategy confers complete protection in lethal models and shows no evidence of interference or disease enhancement when combining two distinct orthoflavivirus antigens [196]. While these approaches are still in preclinical stages, the versatility of the emerging orthoflavivirus vaccine platforms discussed in this review supports their rapid iteration across many orthoflaviviruses. The development of modified-nucleotide saRNA, displaying enhanced antigen expression [210], opens further avenues for the generation of multivalent genetic vaccines against orthoflaviviruses.

Finally, insect-specific flavivirus (ISFV) chimera vaccines have also demonstrated multivalent capability in animal models [292]. Chimeric particles presenting structural antigens from multiple orthoflaviviruses can be produced in mosquito cells, offering pan-orthoflavivirus VLP immunogenicity without risk of vertebrate infection [292].

In summary, while no multivalent orthoflavivirus vaccine is yet licensed, robust platforms, especially VLP-based and nucleic acid technologies, are showing encouraging preclinical results. While these platforms could enable large-scale protection against multiple orthoflaviviruses at once, comprehensive preclinical and clinical studies will be needed to ensure the safety profiles of these vaccine approaches, particularly in relation to ADE.

## 4. Key Attributes of a Robust Orthoflavivirus Vaccine

Without any approved antiviral treatments against orthoflavivirus infections, prevention through mosquito control and vaccination remains our strongest defense. Any orthoflavivirus vaccine should meet a high bar: it should confer durable and strong (ideally sterile) immunity, be robustly safe across diverse populations (including children, pregnant women, and immunocompromised persons) and carry minimal risk of ADE. Ideally, this vaccine should also elicit pan-flavivirus immunity and be deployable in resource-constrained settings where vector-borne threats are most prevalent (Figure 3).

### 4.1. Must-Have Attributes

First, robust and durable vaccine-induced humoral and/or cellular immunity is critical to ensure long-lasting protection against orthoflaviviral diseases. Vaccines such as live-attenuated 17D (for yellow fever) and SA14-14-2 (for Japanese encephalitis) provide long-lasting immunity after a single dose, setting expectations for the rest of the orthoflaviviruses [134,135,293]. Emerging vaccine platforms, from nucleotide-based vaccines to VLPs and vector platforms, will ideally have to meet these benchmarks.

Second, safety and reactogenicity go hand in hand; vaccines need to elicit a protective immune response while mitigating undue discomfort or severe adverse effects. Ensuring this balance is critical for transitioning from clinical approval to widespread public acceptance; this is particularly crucial at a time of increasing vaccine hesitancy. VLPs and vector-based vaccines may be particularly advantageous from this perspective as they preclude risks of reversion or replication, while still promoting robust immune activation through antigen presentation or intrinsic adjuvanticity [214,215,219,220,227,231,232,266,267,274,275,291,294,295,296,297]. Adverse reactions to mRNA and saRNA vaccination caused by transient inflammation, albeit relatively mild for the large majority of patients, will have to be mitigated to avoid public hesitancy toward these life-saving vaccines.

Third, mitigation of ADE is a critical and non-negotiable component of any orthoflavivirus vaccine. Vaccines that induce poorly neutralizing or cross-reactive antibodies, as observed with Dengvaxia, can exacerbate disease upon natural infection [140]. To prevent this, modern candidate designs focus on structural engineering of prM-E, disease-relevant epitope selection, or inclusion of auxiliary antigens such as NS1, with VLPs and mRNA platforms offering versatile and controlled expression to minimize ADE.

Fourth, regulatory rigor and manufacturing reliability cannot be overlooked. Vaccine platforms must navigate rigorous testing to demonstrate consistency, safety, and efficacy, especially in the absence of therapeutic fallbacks. VLPs, mRNA and vector-based platforms offer flexible GMP manufacturing pathways, while more recent platforms like ISFV and saRNA require additional standardization and regulatory validation.

### 4.2. Desirable Attributes

First, protection against infection (i.e., sterilizing immunity), rather than just protection from disease, would be highly valuable for orthoflavivirus vaccines, especially since it could help reduce further viral transmission during outbreaks.

Second, maintaining homogenous and authentic E oligomerization through incorporation onto viral (-like) particles could significantly enhance humoral responses while limiting ADE risks. This could help achieve sterilizing immunity. ISFV chimeras and E-pseudotyped vector platforms, and to a lesser extent VLPs, may be valuable for this purpose through their higher safety profiles relative to live-attenuated vaccines.

Third, multivalent breadth is increasingly recognized as a valuable characteristic. As co-circulation of orthoflavivirus intensifies, and orthoflaviviruses emerge and re-emerge in overlapping geographic areas, a single-agent vaccine targeting multiple orthoflaviviruses would offer significant benefits from a global health management perspective, particularly if it also displays cross-reactive efficacy against non-antigen-matched, related orthoflaviviruses. Critically, multivalence, if adequately designed, could also represent a considerable barrier against ADE. Platforms such as saRNA, VLP-based cocktails, and ISFV chimeras offer scalable solutions for customizable multivalent design, facilitating broader protection. Synergistic, multivalent humoral responses against a specific orthoflavivirus could also help achieve sterilizing immunity in certain infection contexts.

Fourth, logistical practicality strengthens global impact. Cold-chain independence, ease of administration, and rapid scalability represent considerable advantages. Emerging platforms such as saRNA, especially when formulated into thermostable nanoparticles [203], and plant-based VLP [230] production hold promise for deployment in low-resource settings where vector control infrastructure is inconsistent. Logistical practicality can, however, conflict with cost-effectiveness, an equally critical factor for the broad-scale implementation of orthoflavivirus vaccines, particularly in low-income countries.

## 5. Advancing Orthoflavivirus Vaccine Research

The escalating global impact of orthoflaviviruses and the resurgence of pathogens such as YFV, ZIKV, JEV and WNV, highlights a critical gap in our public health defenses. In the absence of antiviral drugs, vaccination remains the cornerstone of effective disease mitigation. Yet, despite multiple available vaccines for some orthoflaviviruses, our arsenal remains narrow and limited by safety issues (i.e., ADE), manufacturing hurdles, and suboptimal immunogenicity/durability and epitope presentations.

Live-attenuated vaccines, such as YFV-17D and JEV SA14-14-2, remain the gold standard for inducing durable humoral and cellular immunity. However, their replication competence presents safety concerns, particularly in immunodeficient individuals [149]. Use of full-length prM-E predisposes them to ADE-related risks [59,60,137,298], particularly in orthoflavivirus-naïve individuals or regions with co-circulating viruses. While inactivated whole-virus vaccines are safer and suitable for vulnerable populations, they often necessitate booster doses and adjuvants to elicit adequate immunity [299,300,301,302,303]. Their inability to induce robust CD8^+^ T-cell responses and reliance on maintaining structural integrity during inactivation complicate their utility [153,165,304,305].

By contrast, structural platforms such as VLPs and ISFV chimera vaccines are prone to present authentic E epitopes while offering higher safety and immunogenic profiles compared to live-attenuated vaccines and inactivated vaccines, respectively. Recent VLP candidates demonstrated potent Th1 immune responses and sterilizing protection in mice without adjuvant use [232,306,307]. Likewise, ISFV chimera vaccines have induced durable, single-dose immunity in multiple orthoflavivirus models [242,243,308,309]. Nucleic acid vaccines advance the immunological toolkit. The genetic versatility of mRNA vaccines facilitates the fine-tuning of antigen design to incorporate prM truncations or enhance E dimer stabilization, while saRNA vaccines expand the immunogenic potential by inducing robust innate activation at low doses; though balancing immune stimulation with IFN suppression remains a key engineering challenge. Although this review primarily focuses on E as the primary antigen for vaccine development, recent findings also suggest that vaccine-induced T-cell responses targeting the orthoflavivirus capsid can protect against orthoflavivirus infection [310]. These results underscore the value of exploring more intensively the potential of E-independent orthoflavivirus vaccines to build next-generation orthoflavivirus vaccines without any risks of ADE.

Ultimately, optimal orthoflavivirus vaccines will have to be structurally precise, immunologically robust, and contextually adaptable to diverse epidemiological landscapes, ideally eliciting durable neutralizing antibodies and T-cell responses without significant safety concerns and risk of ADE. Long-lasting protection against disease will be essential, but an orthoflavivirus vaccine that also confers sterilizing immunity could carry significant advantages by preventing viral transmission from humans to arthropods. Such an attribute could be instrumental for rapidly controlling viral outbreaks. Thermostable formulations and scalable manufacturing will be essential for deployment in resource-limited settings. Platforms such as saRNA nanocarriers, plant-based VLPs, and thermostable mRNA formulations align with these objectives. Of note, our current understanding of orthoflavivirus emergence and evolution suggests that the platforms discussed in this review should also apply to currently neglected and yet-to-be-identified orthoflaviviruses.

Next-generation orthoflavivirus vaccines could be embodied by heterologous prime-boost regimens pairing genetic and structural vaccine platforms. For example, a rapid mRNA or saRNA-based vaccine response encoding for B- and/or T-cell epitopes followed by a VLP or ISFV boost may deliver rapid protection with sustained immunogenicity. Such strategies could mimic the breadth and durability of live-attenuated vaccines while attenuating safety concerns. Combining this approach with multivalent (pan-orthoflavivirus) platforms will be paramount to streamline the clinical management of many orthoflavivirus infections, particularly in areas where many co-circulate.

## 6. Conclusions

In summary, addressing the growing threats from orthoflavivirus require vaccines that combine the immunological depth of live-attenuated platforms with the precision, safety, and scalability of emerging platforms. As global climate change and urbanization continue to expand vector habitats, the development and equitable deployment of such next-generation vaccines become more critical than ever.

## Figures and Tables

**Figure 1 vaccines-13-01015-f001:**
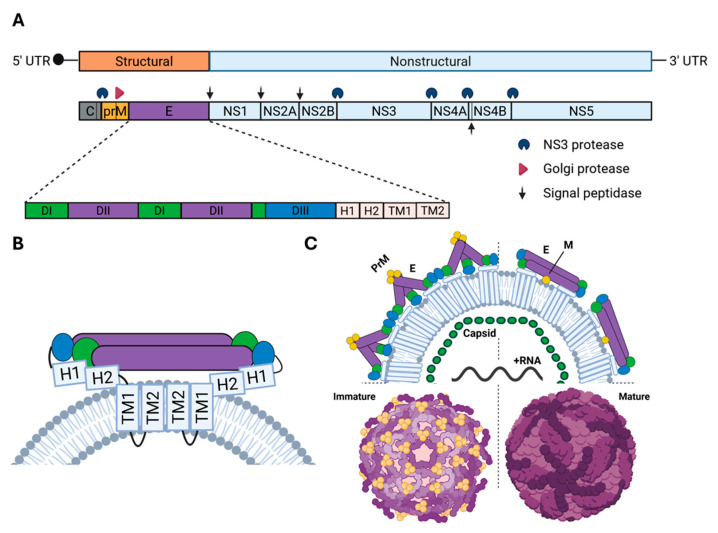
**Generic molecular organization of the orthoflavivirus genome and particle.** (**A**) Orthoflaviviruses encode a single open reading frame (ORF) that is translated in the endoplasmic reticulum (ER) into a large polyprotein. This precursor is co- and post-translationally cleaved by both host and viral proteases to generate ten mature proteins: the three structural proteins (capsid [C], precursor membrane [prM], and envelope [E]) and seven non-structural proteins (NS1–NS5), which together coordinate virion assembly and replication. The E protein, a class II viral fusion glycoprotein, consists of three distinct regions: domain I (EDI, green), the central β-barrel scaffold; domain II (EDII, purple), an elongated finger-like domain that includes the fusion loop; and domain III (EDIII, blue), an immunoglobulin-like domain involved in receptor binding. The ectodomain is followed by a stem region containing two amphipathic α-helices (H1 and H2), which tether the ectodomain to the viral membrane. The protein is anchored within the lipid bilayer via two antiparallel transmembrane helices (TM1 and TM2). (**B**) Schematic illustration of E protein dimers in the prefusion state lying flat along the viral envelope of mature mosquito-borne orthoflavivirus particles under neutral pH conditions in a vertebrate host. (**C**) Illustration of the structural organization of immature (left) and mature (right) mosquito-borne orthoflavivirus particles in a vertebrate host. Immature virions display a spiky surface composed of trimeric prM–E complexes, with the pr peptide shielding the fusion loop of E. In contrast, mature virions exhibit a smooth surface formed by 90 E protein dimers arranged in a herringbone pattern. While mature particles incorporate two membrane proteins, M and E, immature particles contain the uncleaved precursor prM in complex with E.

**Figure 2 vaccines-13-01015-f002:**
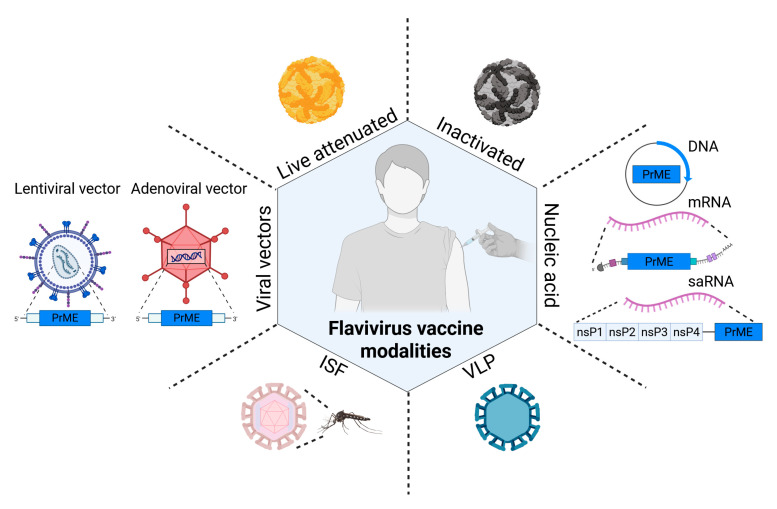
**Orthoflavivirus vaccine platforms.** Schematic representation of major vaccine platforms for orthoflaviviruses, all of which rely on the expression of the structural prM and E proteins to elicit protective immunity. Platforms include: live attenuated vaccines, derived from replication-competent but attenuated viruses; inactivated whole-virus vaccines; nucleic acid vaccines, including DNA, mRNA and self-amplifying RNA (saRNA) vaccines; virus-like particles (VLPs) generated by recombinant expression of prM/E in producer cells; insect-specific flavivirus (ISFV) chimeras, which express prM/E in a non-replicating mosquito-restricted viral backbone; and viral vector-based approaches such as adenoviral or lentiviral vectors to deliver and express prM/E in host cells.

**Figure 3 vaccines-13-01015-f003:**
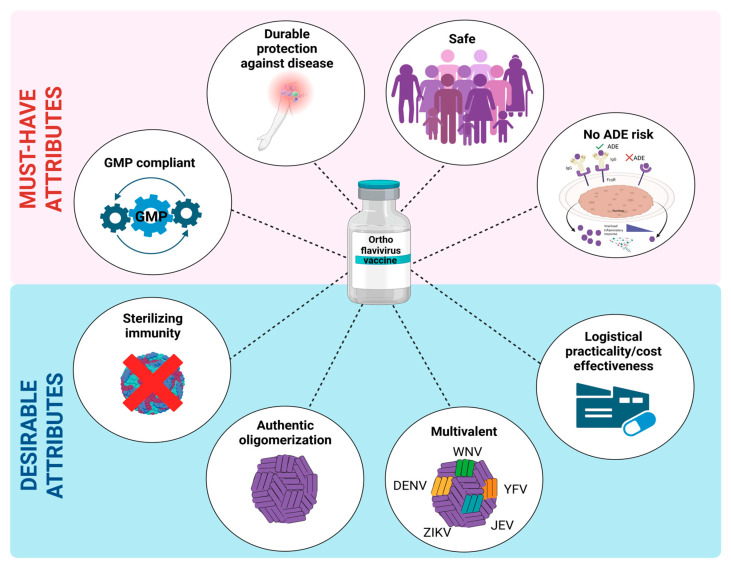
**Key attributes of a robust orthoflavivirus vaccine.** Schematic illustration showing the must-have (red) and desirable (blue) attributes of a robust flavivirus vaccine. Must-have attributes include the induction of durable protective immunity against orthoflaviviral disease, safety across diverse populations, absence of antibody-dependent enhancement (ADE) and Good Manufacturing Practice (GMP) compliance. Desirable attributes include sterilizing immunity, mimicry of authentic E oligomerization and structural epitopes, multivalency driving protection against multiple orthoflaviviruses, and logistical practicality—including scalability for large-scale production, cold-chain independence, and ease of administration—which should be balanced with cost-effectiveness.

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
