# Peer review of "Orthoflavivirus Vaccine Platforms: Current Strategies and Challenges"

_vaccines, 2025, doi:10.3390/vaccines13101015_

Round 1

Reviewer 1 Report

Comments and Suggestions for Authors

Under conflict of interests, the authors report their inventorship on patent applications concerning the use of modified-nucleotide saRNA vaccines against orthoflaviviruses.  This helps explain their approach to their review.  Their expertise and experience can clearly bring value to a review.  As presented, this added value is not evident.  

There are many aspects not covered by the title of the review.  For example: 

  1. Information on tick-borne orthoflaviviruses is scarce despite the fact Crimean-Congo haemorrhagic fever (CCHF) virus is the most lethal (to humans) of all orthoflaviviruses and an emerging threat in Europe (CCHF virus is not mentioned in the review) .  Tick-borne encephalitis (TBE) virus vaccines are routinely used in certain parts of Eurasia (TBE virus vaccines are not mentioned in the review).
  2. The review assumes vaccines are the way forward without considering the growing and promising developments in antiviral therapies.
  3. There is no mention of anti-vector vaccine developments.

If the authors wish to focus on their own area of expertise, the review would benefit from greater comparison of the pros & cons of different vaccine platforms, and look more beyond orthoflaviviruses for insights that can be applied in orthoflaviviruses vaccine development (as the authors indeed do in the case of conventional mRNA vaccines).  What new insights can be gained from AI-driven platforms?

Like all reviews, once written they are out-of-date.  For instance, an update needs to consider implications of a clinical trial showing evidence vaccine-induced T cell responses provide protection without neutralizing antibodies (https://www.nature.com/articles/s41564-024-01903-7.pdf).  

Author Response

Information on tick-borne orthoflaviviruses is scarce despite the fact Crimean-Congo haemorrhagic fever (CCHF) virus is the most lethal (to humans) of all orthoflaviviruses and an emerging threat in Europe (CCHF virus is not mentioned in the review) . 

 While we acknowledge the growing threat posed by CCHFV in Europe and elsewhere, CCHFV is classified as a nairovirus, not an orthoflavivirus. As such, its exclusion was intentional.

Tick-borne encephalitis (TBE) virus vaccines are routinely used in certain parts of Eurasia (TBE virus vaccines are not mentioned in the review).

We thank the reviewer for bringing this important point. We have included a dedicated paragraph discussing licensed TBEV vaccines (section 2.2) and their use in endemic areas.

The review assumes vaccines are the way forward without considering the growing and promising developments in antiviral therapies.

We agree that antiviral development is an essential and complementary strategy to vaccines. However, the scope of this review is defined explicitly around vaccine development and their preventive potential. At no point do we claim that vaccine development is more critical than antiviral therapies in combating flavivirus infection. Nevertheless, we have added the following in our introduction “there is a pressing need to develop effective therapeutic and prophylactic strategies to reduce their global health impact”.

There is no mention of anti-vector vaccine developments.

We appreciate this suggestion. In the revised manuscript, we now briefly discuss anti-vector approaches, including anti-tick and mosquito salivary protein vaccines and transmission-blocking strategies (Section 2.7).

If the authors wish to focus on their own area of expertise, the review would benefit from greater comparison of the pros & cons of different vaccine platforms, and look more beyond orthoflaviviruses for insights that can be applied in orthoflaviviruses vaccine development (as the authors indeed do in the case of conventional mRNA vaccines). 

We thank this reviewer for their suggestions. To goal of this review is to provide a brief overview of the different flavivirus vaccine platforms available, and their respective pros and cons (see Table 1). Expanding such comparative analysis, and looking beyond orthoflaviviruses for strategies, would significantly extend the length of our manuscript and change the overall “snapshot” nature of our review (we have updated our abstract to further stress this point, from “brief overview” to “snapshot”). We would be happy to expand this further in a dedicated review or follow-up study if appropriate.

What new insights can be gained from AI-driven platforms?

We greatly appreciate your remark regarding AI’s potential in advancing flavivirus vaccine design. At this stage, we believe AI integration within flavivirus research (and vaccine research in general) is still in its early stages. Whether AI models can be expanded from current predictive models of optimal vaccine-induced immune responses to predictive models of innovative, immunogenic platforms (immunogen and delivery) remains to be experimentally validated. It is also unclear how the high energy demands of AI models will be met as the usage of this technology continues to expand. Given the “snapshot” nature of our review and the immaturity of AI models in vaccine research, we feel it is still premature to discuss this topic in the context of our manuscript. We look forward to revisiting this fascinating topic in the future.

Like all reviews, once written they are out-of-date.  For instance, an update needs to consider implications of a clinical trial showing evidence vaccine-induced T cell responses provide protection without neutralizing antibodies (https://www.nature.com/articles/s41564-024-01903-7.pdf).  

We have added this reference in section 5.

Reviewer 2 Report

Comments and Suggestions for Authors

In this review, the authors describe the various platforms for B-cell based flavivirus vaccines, the pros and cons of each platform and list the characteristics of the ideal flavivirus vaccine.

This work is significant and will be useful to scientists in the field. However, the authors should reconsider or revise parts of the text as discussed below.

Major concerns:

  • Add references in Table 1
  • The title mentions “emerging solutions” but it is not clear whether the text really offers solutions for the complex problem of generating a successful orthoflavivirus vaccine, as also mentioned in Lines 24-28.
  • Along the same lines, it is not clear whether the “ideal orthoflavivirus vaccine” must check all points listed. A practically feasible and successful vaccine is unlikely to achieve all listed goals. Perhaps this section can be written to leave room for not strictly adhering to a rigid list, can have a list of must-have qualities and desirable qualities to start with.

Other concerns:

  • Lines 44-47: it would be helpful to include the geographical areas where the stated viruses are emerging.
  • Line 135: “virus-infectious” or infectious virus? It might be worth mentioning the role of prM in an infectious particle, and how this differs in mosquito-borne vs tick-borne viruses (Holoubek et al 2025: https://doi.org/10.1038/s41467-025-62750-6)

Author Response

Add references in Table 1

We have now incorporated appropriate references for each vaccine platform listed in Table 1, aligning with the most recent and relevant literature.

The title mentions “emerging solutions” but it is not clear whether the text really offers solutions for the complex problem of generating a successful orthoflavivirus vaccine, as also mentioned in Lines 24-28.

We agree that the title should accurately reflect the scope. We have revised the title to:
“Orthoflavivirus Vaccine Platforms: Current Strategies and Challenges”. This change better reflects our aim to describe promising platforms rather than definitive solutions.

Along the same lines, it is not clear whether the “ideal orthoflavivirus vaccine” must check all points listed. A practically feasible and successful vaccine is unlikely to achieve all listed goals. Perhaps this section can be written to leave room for not strictly adhering to a rigid list, can have a list of must-have qualities and desirable qualities to start with.

We thank this reviewer for their comment. We agree with their comment and have reorganized section 4 accordingly. We have notably segregated the qualities of a robust orthoflavivirus vaccines between “must-have” and “desirable” as suggested. We have also removed the concept of “ideal” flavivirus vaccine.

Lines 44-47: it would be helpful to include the geographical areas where the stated viruses are emerging.

We have now added geographical context to this sentence.

Line 135: “virus-infectious” or infectious virus? It might be worth mentioning the role of prM in an infectious particle, and how this differs in mosquito-borne vs tick-borne viruses (Holoubek et al 2025: https://doi.org/10.1038/s41467-025-62750-6)

This sentence now reads “Upon exocytosis at neutral pH, a localized conformational change within E DI re-leases the pr peptide, yielding mature, infectious virions that display 90 E homodimers organized in a smooth conformation”.

We are also discussing differences in particle maturation between mosquito-borne and tick-borne flaviviruses more extensively, and discuss findings from Holoubek et al. (2025). We notably write “Tick-borne orthoflavivirus immature particles, unlike mosquito-borne orthoflaviviruses, also remain fully infectious because of the irreversible exposure of the FCS at neutral pH; which enables furin-mediated cleavage of prM at the surface of target cells upon viral entry”.

Reviewer 3 Report

Comments and Suggestions for Authors

The full report is attached.

Comments on the Quality of English Language

The manuscript is written in generally clear and readable English, but it would benefit significantly from professional language editing. While the scientific ideas are communicated successfully, there are recurrent issues with syntax, redundancy, wordiness, and flow that occasionally obscure meaning or reduce the precision of the argument. The tone is mostly appropriate for a scientific review, but at times the style becomes overly conversational or ambiguous. Several sentences rephrase the same point using slightly different wording, which interrupts flow and makes the text longer than necessary. Example: Expressions like “Orthoflaviviruses, such as those in the DENV, ZIKV, and YFV groups” are repeated across sections without adding new nuance.
- Phrases such as “This could potentially lead to situations where” or “The reason for this might be due to the fact that” can be simplified to improve clarity. Use more concise structures (e.g., “This may lead to…”).
- Inconsistent use of “flavivirus” and “orthoflavivirus” without clarification may confuse readers. Some technical terms are introduced without definition or are used inconsistently (e.g., “immune imprinting,” “cross-protection,” “antigenic sin”).
- Occasional misuse of definite and indefinite articles, as well as missing prepositions (e.g., “leads production of” instead of “leads to the production of”).
- Several sections lack smooth transitions, causing abrupt shifts in topics. Logical connectors (e.g., “however,” “in contrast,” “additionally”) are sometimes missing or misused.
- The manuscript occasionally slips into a less formal register, using phrases that sound conversational rather than precise or technical. Example: “This makes things more complicated” could be revised to “This adds complexity to vaccine design.”

But some positive points also can be highlighted:  the manuscript avoids slang or overly casual language. The authors generally maintain a professional tone, especially in the discussion of vaccine platforms. Spelling and punctuation are consistent, with only a few minor typographical errors noted. My recommendation is minor to moderate language editing required, that the authors submit the manuscript for professional English language editing, especially to remove redundancy; improve flow and transitions; refine scientific tone and phrasing; clarify key terms and ensure consistency across sections. This will enhance readability and ensure that the manuscript reaches its full potential in terms of scientific communication and impact.

Author Response

We thank this reviewer for the detailed and constructive feedback. We thoroughly went through our manuscript to remove redundancy, enhance logical flow, and refine the scientific tone. We systematically revised repetitive expressions, and we have harmonized the use of “flavivirus” versus “orthoflavivirus”. Instances of article misuse and missing prepositions have been meticulously corrected. We have improved transitions, and conversational expressions have been revised to more formal alternatives.

Reviewer 4 Report

Comments and Suggestions for Authors

The key summarized tables or figures are absent in this review.
The summarized schematic of a comparable of the viral genomes and constructed proteins in the manuscript would be recommended.
The summary of the targets of different flavivirus vaccine and the developed company should also be listed. 
The vaccine current status and analysis of Dengue virus (DENV), Zika virus (ZIKV), yellow fever virus (YFV) and West Nile virus (WNV) should be summarized in a table individually because of their different challenge.
Some minor issues should be careful corrected
In line 134, pr is a irregular word,
Upon exocytosis at neutral pH, pr is released, and the mature virion displays 90 E 134
The following sub-titles should add the vaccine.
2.3.2. Self-Amplifying RNA (saRNA)
2.4. Virus-like particles (VLPs)

Author Response

The key summarized tables or figures are absent in this review. The summarized schematic of a comparable of the viral genomes and constructed proteins in the manuscript would be recommended.

Our manuscript originally included a summary Table 1, which we have expanded in this revised version. However, we have now added three new figures, including a schematic of the different orthoflavivirus vaccine platforms being discussed.

The summary of the targets of different flavivirus vaccine and the developed company should also be listed. The vaccine current status and analysis of Dengue virus (DENV), Zika virus (ZIKV), yellow fever virus (YFV) and West Nile virus (WNV) should be summarized in a table individually because of their different challenge.

Since our review aims to provide a brief overview of the current orthoflavivirus vaccine, with a focus on the platforms rather than specific vaccines, we believe including such details would exceed the scope of our review and its “snapshot” nature. We may consider these inclusions for a standalone review in the future.

In line 134, pr is a irregular word, Upon exocytosis at neutral pH, pr is released, and the mature virion displays 90 E 134.

We have clarified our sentence as “a localized conformational change within E DI releases the pr peptide”.

The following sub-titles should add the vaccine: 2.3.2. Self-Amplifying RNA (saRNA), 2.4. Virus-like particles (VLPs).

Subtitles for the relevant subsections now include the words “vaccine”. Specifically, “2.3.2. Self‑Amplifying RNA (saRNA) Vaccine” and “2.4. Virus‑Like Particle (VLP) Vaccine”.

Round 2

Reviewer 1 Report

Comments and Suggestions for Authors

The new title of the review more accurately reflects the contents and there is now some recognition of tick-borne orthoflaviviruses and the well-established TBE virus vaccine (sorry about the inappropriate reference to CCHF nairovirus). 

There are still a number of important points not fully addressed in the revised manuscript. 

  1. The review focuses primarily on strategies aimed at inducing neutralizing antibodies targeting the E protein.  However, cited reference [294] provides evidence of vaccine-induced protection without neutralizing antibodies in humans.  The potential significance of this observation needs considering in terms of strategies and challenges.
  2. The challenge model also needs considering in terms of vaccine platforms.  Most challenge studies involve syringe inoculation of model animals or humans whereas natural infection is via an infected mosquito/tick which may give a different outcome.
  3. Anti-vector vaccines offer the potential for protection against all the pathogens transmitted by a specific arthropod vector, whether known or unknown.  This obvious attraction of an anti-vector vaccine strategy should be mentioned as well as clinical trials with anti-mosquito vaccines (eg. https://doi.org/10.1016/S0140-6736(20)31048-5).
  4. Arguably, the most important consideration for an orthoflavivirus vaccine is cost effectiveness, which is not really discussed.
  5. There is no mention of tick vectors/tick-borne orthoflaviviruses in the Abstract.
  6. Table 1 (labelled ‘Tables’) refers only to mosquito-borne viruses.
  7. Anti-vector vaccines do not necessarily block pathogen transmission.  For example, cited reference [265] showed that protection against lethal challenge with TBE virus was dependent on neutralising antibodies to TBE virus i.e. the virus was transmitted but its lethality was attenuated by the anti-tick immune response.
  8. Reference [266] is cited as an anti-vector (mosquito) vaccine whereas it is an anti-viral vaccine delivered by mosquito.
  9. Fig. 1 does not distinguish between features of mosquito- and tick-borne viruses.  Presumably, the depiction relates to orthoflavivirus structure in the vertebrate host whereas the virus transmitted in mosquito/tick saliva may be in an alkaline environment and/or within microvesicles and have a slightly different structure.
  10. Fig. 3 ‘Desired properties…’. In fact, to be approved for human use, many properties are essential and the overarching need is to be cost effective.

Author Response

The review focuses primarily on strategies aimed at inducing neutralizing antibodies targeting the E protein.  However, cited reference [294] provides evidence of vaccine-induced protection without neutralizing antibodies in humans.  The potential significance of this observation needs considering in terms of strategies and challenges.

We acknowledge that T cell responses can drive protection even in the absence of neutralizing antibodies; however, our review focuses on Envelope (E) protein-based vaccines, as stated in our abstract and section 2, consistently with evidence that neutralizing antibodies targeting the E protein remain the most consistent and validated correlate of protection against orthoflaviviruses.

The challenge model also needs considering in terms of vaccine platforms.  Most challenge studies involve syringe inoculation of model animals or humans whereas natural infection is via an infected mosquito/tick which may give a different outcome.

We appreciate this distinction. However, in this review, our primary focus is on vaccine platforms and antigen design, rather than on preclinical models of vaccination & challenge. We believe that while non-physiologically relevant challenge methodologies could confound vaccine efficacy studies, this point is beyond the scope of our review.

Anti-vector vaccines offer the potential for protection against all the pathogens transmitted by a specific arthropod vector, whether known or unknown.  This obvious attraction of an anti-vector vaccine strategy should be mentioned as well as clinical trials with anti-mosquito vaccines (eg. https://doi.org/10.1016/S0140-6736(20)31048-5).

We thank the reviewer for this suggestion. We have commented this in the Anti-vector vaccines section as follows: “A notable advantage of such a strategy is its ability to target any pathogens transmitted by arthropods, whether known or unknown.” We have also cited the suggested clinical trial (DOI: 10.1016/S0140-6736(20)31048-5) as an example of an anti-mosquito vaccine entering clinical testing.

Arguably, the most important consideration for an orthoflavivirus vaccine is cost effectiveness, which is not really discussed.

We agree that cost-effectiveness is a key factor for large scale implementation of vaccines. We have made mention of this point in Section 4.2.

There is no mention of tick vectors/tick-borne orthoflaviviruses in the Abstract.

We thank the reviewer for this important oversight. We have revised our Abstract accordingly.

Table 1 (labelled ‘Tables’) refers only to mosquito-borne viruses.

We thank the reviewer for this observation. In response, we have updated Table 1 to include a TBEV vaccine (Ticovac) as an example. We would also like to clarify that the references in this table already covered TBEV vaccines.

Anti-vector vaccines do not necessarily block pathogen transmission.  For example, cited reference [265] showed that protection against lethal challenge with TBE virus was dependent on neutralising antibodies to TBE virus i.e. the virus was transmitted but its lethality was attenuated by the anti-tick immune response.

This point has been clarified as follows: “Mouse immunization with specific modalities of the 64TRP vaccine reduced viral transmission from infected tick vectors to mice and vice versa, and decreased the incidence of fatal infection to a similar extent as a licensed TBEV vaccine”.

Reference [266] is cited as an anti-vector (mosquito) vaccine whereas it is an anti-viral vaccine delivered by mosquito.

We thank the reviewer for identifying this error. We have corrected the reference citation and added accurate references describing vaccines targeting mosquito salivary proteins.

Fig. 1 does not distinguish between features of mosquito- and tick-borne viruses.  Presumably, the depiction relates to orthoflavivirus structure in the vertebrate host whereas the virus transmitted in mosquito/tick saliva may be in an alkaline environment and/or within microvesicles and have a slightly different structure.

We have added a clarifying statement in the legend of Fig. 1 specifying that the depicted virion structure refers to mosquito-borne orthoflavivirus in the vertebrate host environment.

Fig. 3 ‘Desired properties…’. In fact, to be approved for human use, many properties are essential and the overarching need is to be cost effective.

We agree with the reviewer’s suggestion. In Fig. 3, we have updated the “Logistical Practicality” bubble to explicitly include “Cost-effectiveness” as another desirable feature of the orthoflavivirus vaccine. This change is also reflected in the accompanying text.

Reviewer 2 Report

Comments and Suggestions for Authors

The authors have addressed primary concerns to this reviewer’s satisfaction and significantly improved the review, which would be immensely helpful for researchers in the field.

However, please correct or address the following comments prior to publication.

  1. Table 1 references: Assuming that the references are for the platform and not the virus (or is it for both?), please double check that all references are indeed added to the correct row.

  1. Lines 114-117: The reference seems to not quite support the statement, especially that “temperate regions could soon experience recurring outbreaks”.

  1. Upon exocytosis at neutral pH, a localized conformational change within E DI re-leases the pr peptide, yielding mature, infectious virions that display 90 E homodimers organized in a smooth conformation”

To the best of this reviewer’s understanding, the maturation of the particle with the 90 E homodimers displayed in a smooth architecture is not contingent on pr release, as it seems to imply from the way the sentence above is structured. In other words, pr remains associated with the particle, even when the virion has acquired the smooth configuration.

  1. For Table 1 headings for different columns, please consider updating them to: Platform; Examples; Advantages; Disadvantages; Development Stage; References.

  1. Typos: line 310

  1. Figure3: “Desirable features” in the figure should match the color code mentioned in the legend (should be blue in Figure).

  1. Figure 2: Adding written labels for the sub-types of vaccines in each category within the figure might be useful for readers. For instance, in the nucleic acid section, mRNA and saRNA vaccines can be listed. Hence all info comes across from the figure itself. Similarly, SVP particle vaccines can be added too.

  1. Figure 1. Under the schematic for the organization of polyprotein (panel A), adding the polyprotein start (for Capsid) and end (NS5) numbers would be helpful for readers.

Author Response

Table 1 references: Assuming that the references are for the platform and not the virus (or is it for both?), please double check that all references are indeed added to the correct row.

We have carefully reviewed Table 1 and confirm that the references correspond primarily to the platform they are associated with.

Lines 114-117: The reference seems to not quite support the statement, especially that “temperate regions could soon experience recurring outbreaks”.

We agree and have replaced the original citation with a more appropriate reference supporting this statement.

“Upon exocytosis at neutral pH, a localized conformational change within E DI re-leases the pr peptide, yielding mature, infectious virions that display 90 E homodimers organized in a smooth conformation”

To the best of this reviewer’s understanding, the maturation of the particle with the 90 E homodimers displayed in a smooth architecture is not contingent on pr release, as it seems to imply from the way the sentence above is structured. In other words, pr remains associated with the particle, even when the virion has acquired the smooth configuration.

The reviewer is correct. We have revised the sentence as follows: “Particle migration to the trans-Golgi network (TGN), a mildly acidic environment, rear-ranges E-prM trimers to form particles displaying 90 E homodimers organized in a smooth conformation, simultaneously exposing a furin cleavage site (FCS) within prM[95-99]. Furin, a resident TGN protein, then cleaves prM into M and a pr peptide[97,98]. An N-terminal, “globular,” pr peptide remains associated with E, shielding the E fusion loop domain and preventing premature fusogenic rearrangement of the E dimers[96-98,100]; While the C-terminal, membrane-anchored part of prM (i.e., M) sits below the E dimers. Upon exocytosis at neutral pH, a localized conformational change within E DI re-leases the pr peptide, yielding mature infectious virions [100-103]”.

For Table 1 headings for different columns, please consider updating them to: Platform; Examples; Advantages; Disadvantages; Development Stage; References.

We thank the reviewer for the suggestion, we have updated the column headings in Table 1 as recommended.

Typos: line 310

 The typos on line 310 have been corrected.

Figure3: “Desirable features” in the figure should match the color code mentioned in the legend (should be blue in Figure).

We thank the reviewer for pointing this out. We have revised Figure 3 to ensure that all visual elements, including the "Desirable Features" section, follow the correct color scheme as indicated in the legend for better clarity.

Figure 2: Adding written labels for the sub-types of vaccines in each category within the figure might be useful for readers. For instance, in the nucleic acid section, mRNA and saRNA vaccines can be listed. Hence all info comes across from the figure itself. Similarly, SVP particle vaccines can be added too.

We have modified Figure 2 to include sub-category labels for each vaccine platform, such as “DNA”, “mRNA” and “saRNA” under the nucleic acid section.

Figure 1. Under the schematic for the organization of polyprotein (panel A), adding the polyprotein start (for Capsid) and end (NS5) numbers would be helpful for readers.

We appreciate the suggestion. However, we have opted not to include specific amino acid positions in Figure 1A, as the schematic is intended to represent the generic organization of the orthoflavivirus polyprotein. Given the variability in polyprotein length among different orthoflaviviruses, adding precise numbering could be misleading.

Reviewer 3 Report

Comments and Suggestions for Authors

The full report is attached.

Author Response

While the paper does mention lesser-known viruses (e.g., ROCV, ILHV, CPCV), these still receive limited depth compared to dengue or yellow fever. Given the article's stated goal to highlight “emerging solutions,” more emphasis on challenges in vaccines for these neglected viruses would enhance the review’s originality and utility.

We appreciate this observation. Our review aims to highlight vaccine platform versatility rather than pathogen-specific solutions. We have clarified this point in section 5, emphasizing that “our current understanding of orthoflavivirus emergence and evolution suggests that the platforms discussed in this review should also be applicable to currently neglected and yet-to-be-identified orthoflaviviruses.”

There is no dedicated discussion on the peculiarities of orthoflavivirus immunity or vaccine response in infants or children, key target populations for many of these viruses (especially DENV and YFV). Even a short paragraph would improve the clinical applicability of the review.

We agree that pediatric populations are important targets for orthoflavivirus vaccines. However, we respectfully note that our review is focused on orthoflavivirus vaccine platforms rather than population-specific immunology. Given space constraints and the breadth of the topic, a dedicated discussion on pediatric immunology is beyond the current scope. That said, we do mention in section 1.1 that young children are a key population to target for vaccination.

While the consistent usage is appreciated, the authors could include a brief explanation of the taxonomic update (e.g., when the ICTV officially adopted "Orthoflavivirus", why the change was made, and its implications) in the introduction. This helps orient readers who are more familiar with the traditional term "flavivirus."

We appreciate this suggestion. However, we chose not to include a detailed taxonomic note as the review focuses on vaccine development and not viral classification. A discussion of ICTV nomenclature changes, while interesting, would fall outside the thematic scope.

L93–L95: The sentence discussing global outbreaks lacks citation. Consider referencing

WHO or CDC data for validation.

We have updated the sentence and added multiple citations.

L304–L312: In the discussion of mRNA vaccines, include a reference to limitations such

as cold-chain dependency and potential lipid nanoparticle reactogenicity.

We appreciate the reviewer’s suggestion. We would like to clarify that these aspects were already referenced in the original version of the manuscript. Nonetheless, in response to the reviewer’s comment, we have now added additional references related to LNP reactogenicity.

  1. L440: When discussing novel antigen design (e.g., mosaic nanoparticles), it would be

helpful to include examples of existing studies or trials.

We thank the reviewer for bringing this to our attention. Upon double-checking, we were not able to locate the section associated with this reviewer’s comment in line 440. We also do not make mention of mosaic nanoparticles.

Improved significantly the grammar/tone. Some minor polishing could still enhance flow in longer sentences, but overall, the language is now acceptable for publication. This manuscript is now significantly improved and demonstrates value to the Vaccines readership. It offers an accessible yet rigorous review of orthoflavivirus vaccine development. Pending minor adjustments (especially the addition of a table and more attention to neglected viruses) I believe the paper is suitable for publication.

We thank this reviewer for their valuable suggestions to improve our manuscript.

Reviewer 4 Report

Comments and Suggestions for Authors

The revised version has a previous improved.

Author Response

The revised version has a previous improved.

Thank you!

Round 3

Reviewer 1 Report

Comments and Suggestions for Authors

Thank you for addressing my comments.